# Dietary and Biological Assessment of the Omega-3 Status of Collegiate Athletes: A Cross-Sectional Analysis

Peter P. Ritz[1], Mark B. Rogers[1,2], Jennifer S. Zabinsky[1,3], Valisa E. Hedrick[3], John A. Rockwell[4], Ernest G. Rimer[5,6], Samantha B. Kostelnik[3], Matthew W. Hulver[3,7,8], Michelle S. Rockwell [3,7]*

1 Athletics Department, Virginia Tech, Blacksburg, VA, United States of America, 2 Edward Via College of Osteopathic Medicine, Blacksburg, VA, United States of America, 3 Department of Human Nutrition, Foods and Exercise, Virginia Tech, Blacksburg, VA, United States of America, 4 Department of Medicine, Carilion Clinic, Roanoke, VA, United States of America, 5 Department of Exercise & Sport Science, College of Health, University of Utah, Salt Lake City, UT, United States of America, 6 Athletics Department, University of Utah, Salt Lake City, UT, United States of America, 7 Center for Transformative Research on Health Behaviors, Fralin Biomedical Research Institute Roanoke, Roanoke, VA, United States of America, 8 Fralin Life Sciences Institute, Virginia Tech, Blacksburg, VA, United States of America

* msrock@vt.edu

**Data Availability Statement:** All relevant data are within the paper.

**Funding:** This research was supported by the Collegiate and Professional Sports Dietitians

## Abstract

Omega-3 fatty acids (ω-3 FA) are associated with cardiovascular health, brain function, reduction of inflammation, and several other physiological roles of importance to competitive athletes. The ω-3 FA status of National Collegiate Athletic Association (NCAA) Division I athletes has not been well-described. The purpose of this study was to evaluate the ω-3 FA status of NCAA Division I athletes using dietary and biological assessment methodology. Athletes from nine NCAA Division I institutions from throughout the U.S. (n = 1,528, 51% male, 34 sports represented, 19.9 ± 1.4 years of age) completed a food frequency questionnaire (FFQ) to assess ω-3 FA from diet and supplements. Omega-3 Index (O3i) was evaluated in a sub-set of these participants (n = 298, 55% male, 21 sports represented, 20.0 ± 1.3 years of age) using dried blood spot sampling. Only 6% (n = 93) of athletes achieved the Academy of Nutrition & Dietetics' recommendation to consume 500 mg DHA+EPA per day. Use of ω-3 FA supplements was reported by 15% (n = 229) of participants. O3i was 4.33 ± 0.81%, with no participants meeting the O3i benchmark of 8% associated with the lowest risk of cardiovascular disease. Every additional weekly serving of fish or seafood was associated with an absolute O3i increase of 0.27%. Overall, sub-optimal ω-3 FA status was observed among a large, geographically diverse group of male and female NCAA Division I athletes. These findings may inform interventions aimed at improving ω-3 FA status of collegiate athletes. Further research on athlete-specific ω-3 FA requirements is needed.

## Introduction

Omega-3 polyunsaturated fatty acids (ω-3 FA), namely long-chain eicosapentaenoic acid (EPA) and docosahexaenoic acid (DHA), serve as structural components within phospholipid

Association Research Award (www.sportsrd.org) which was awarded to first author Peter P. Ritz in 2018. The funders had no role in study design, data collection and analysis, decision to publish, or preparation of the manuscript.

**Competing interests:** The authors have declared that no competing interests exist.

cell membranes. These ω-3 FA have also been shown to play important physiological roles among the cardiovascular [1–6], nervous [7–13], and skeletal muscle systems [14–18], and in the body's inflammatory response [19–26]. In athletes, ω-3 FA have been associated with the management of exercise-induced oxidative stress [19, 20, 23–25], delayed onset muscle soreness [21, 22, 25, 26], oxygen efficiency during aerobic exercise [2], anaerobic endurance capacity [3], and skeletal muscle health [14–18]. The neuroprotective role of DHA as related to concussion and traumatic brain injury (TBI) risk has also been explored, with particular application to American football athletes [8–13].

As essential fats, EPA and DHA must be obtained exogenously because the human body has limited ability to synthesize these ω-3 FA from precursor ω-3 FA alpha-linolenic acid (ALA) [27]. Fish and seafood are the richest sources of ω-3 FA but there is wide variation in the EPA and DHA content of these foods, ranging from 100 mg to 2500 mg of combined EPA + DHA per three ounce serving [28–32]. Some of the richest food sources, such as salmon, may have up to ten times the EPA and DHA content compared with less rich sources like shrimp and tilapia [28–32]. Also of note, frequent consumption of some common sources of ω-3 FA may risk overexposure to mercury [28].

While there is currently no consensus for ω-3 FA dietary recommendations and athlete-specific recommendations have not been established [33–36], the Academy of Nutrition and Dietetics recommends consumption of 500 mg EPA + DHA daily, estimated to be accomplished via eating at least two servings of fish each week [33, 34]. To meet ALA needs, the National Academy of Medicine (formerly Institute of Medicine) recommends consumption of at least 1.6 g in adult men and 1.1 g daily in adult women [35]. Low ω-3 FA intake appears to be prevalent within the general population of North America [28, 29, 37, 38]. Reports of athletes' ω-3 FA intake are minimal to date, but Wilson and Madrigal [39] observed daily intakes of EPA and DHA below 100 mg in a group of 58 National Collegiate Athletics Association (NCAA) Division I collegiate athletes. Little information is available at present with regard to athletes' habitual use of ω-3 FA supplements.

In addition to ω-3 FA intake, ω-3 FA status may be evaluated using the Omega-3 Index (O3i), which reflects the sum of EPA and DHA in erythrocyte membranes as a percentage of total erythrocyte fatty acids [4]. Compared to other methods, O3i requires a minimum amount of blood (i.e., finger stick blood sample), has a low biological variability [40], is less affected by acute feedings to better reflect long-term ω-3 FA intake [41], and has been shown to correspond with ω-3 FA concentrations in the heart, brain, and a variety of other tissues [42, 43]. An O3i <4% has been associated with the highest risk for the development of cardiovascular disease; whereas, 4–8% is considered moderate risk and ≥8% is the lowest risk [4–6]. Recently, an average O3i of 4.4% was observed among collegiate football athletes at four U.S. universities [44]. A large scale assessment of O3i in collegiate athletes has not been described in the published literature, to our knowledge.

Prior to 2019, the NCAA classified ω-3 FA supplements as "impermissible", which prevented athletic departments from purchasing such supplements for athletes unless prescribed by a team physician [45]. However, advocacy from member institutions facilitated a 2019 NCAA legislation change, reclassifying ω-3 FA supplements as permissible for athletic departments to purchase and provide for athletes [46]. As a result of this rule change, interest in and availability of ω-3 FA supplements has risen. In order to better inform recommendations and ultimately nutrition interventions, a better understanding of athletes' ω-3 FA status is needed. Thus, the purpose of this study was to assess the ω-3 FA intake and O3i of male and female NCAA Division I athletes who participate in a variety of sports.

## Methods

### Study design

A multi-site, cross-sectional study was designed to assess the ω-3 FA dietary intake, ω-3 FA supplement use, and O3i of NCAA Division I athletes. These assessments were carried out during the 2018–2019 academic year.

### Participants

In an effort to recruit a geographically diverse subject pool, the research team solicited volunteer research collaborators (registered dietitians or sports performance staff employed by their respective athletics programs) from the NCAA Division I Power 5 institutional membership (Atlantic Coast Conference, Big Ten Conference, Big Twelve Conference, Pacific-12 Conference, and Southeastern Conference). Research collaborators provided oversite for on-site data collection (in conjunction with the primary research team) and served as liaisons to the primary research team. One institution from each of eight regions throughout the U.S. was accepted (on a first come-first served basis), with the exception of the Northeast from which two were ultimately accepted since one became unable to complete blood measurements after agreeing to participate. The nine participating institutions represented the following states: California, Georgia, Illinois, Nebraska, Oregon, Pennsylvania, Texas, Utah and Virginia. Participating institutions were assured protection of program identity at the level of state.

Research collaborators posted flyers in common athlete facilities and sent email and/or text messages to all athletes within the institution, inviting them to participate in a research study aimed at assessing dietary and supplemental ω-3 FA intake. Male and female athletes who were over the age of 18 years and on a current roster for any NCAA Division I sport at one of the participating institutions were eligible to participate. Although there was some variation in sports offered at the participating institutions, all nine included men's baseball, men's and women's basketball, men's and women's cross country, men's football, men's and women's golf, women's gymnastics, women's soccer, women's softball, men's and women's swimming & diving, men's and women's track & field, and women's volleyball.

### Omega-3 dietary assessment

A twenty six-item food frequency questionnaire (FFQ) validated to assess ω-3 FA dietary intake [39, 47] was administered to eligible participants electronically using Qualtrics (version XM, Provo, Utah, U.S.). Consent to participate was inferred by completion of the FFQ. The FFQ was modified to include demographic characteristics of participants (sex, age, academic year, and sport) and ω-3 FA supplement use. Within the FFQ, participants reported the frequency of consumption and average portion size for an extensive list of ω-3 FA food sources including fish, shellfish, walnuts, canola oil, flaxseed, flaxseed oil, and cod liver oil. For participants who indicated that they consumed ω-3 FA supplements, information about brand, form, dosage, and frequency taken was requested.

The FFQ results were compiled and analyzed using methodology outlined by Sublette et al [47]. Previously published databases [30–32] were used as a reference for ω-3 FA content of foods consumed based on source and portion size reported. Responses with more than one unanswered question were omitted from analyses.

### Blood fatty acid analysis

Following completion of the dietary assessment portion of the study, participants were offered the opportunity to volunteer for a second portion of the study: analysis of blood fatty acids. A

single drop of whole blood was collected from the index or middle finger and applied to a blood spot card pre-treated with an antioxidant cocktail. Blood collection was carried out by a registered dietitian or certified athletic trainer in an athletic training room or sports medicine facility. Samples were shipped to a central laboratory (OmegaQuant, Sioux Falls, South Dakota, U.S.) within fourteen days for analysis of fatty acids and calculation of the O3i using gas chromatography. This methodology was described in detail by Harris and Polreis [48].

## Statistical analysis

Data were analyzed using IBM Statistical Package for the Social Sciences (SPSS) version 26. Descriptive statistics are expressed as means and standard deviations for continuous data, and frequencies and percentages for categorical data. Data were tested for normality using the Shapiro-Wilk test. Differences in outcomes between demographic groups were calculated using analysis of variance (ANOVA) or chi-square tests. Relationships between diet and blood variables were analyzed using Pearson's correlations. Multiple regression analysis was used to assess the effects of diet on O3i with adjustment for demographic covariates including institution, sex, age, class year, and sport. Football vs non-football sport comparisons were made in order to compare results to existing literature, and because football programs may have different resources and athlete characteristics as compared with other sports. Significance was set at a level of $p < 0.05$.

## Ethical considerations

This study was approved by the Institutional Review Board of Virginia Tech (IRB# 18–606) and respective institutional research review committees. Consent for the dietary assessment portion of the study was inferred based on voluntary completion. Written and informed consent was provided by participants before starting the blood fatty acid portion of the study.

## Results

In all, 1562 participants completed the dietary assessment portion of the study and 301 completed the blood analysis portion. Thirty-four FFQs were incomplete and thus, excluded from analysis. We also excluded 3 participants from the blood analysis since they did not have complete FFQ submissions. Overall, a total of 1528 participants (51% males) were included in dietary assessment analyses, and 298 (55% males) in the blood analysis portion. Descriptive characteristics of participants are shown in Table 1. Participants represented 15 different male sports and 19 different female sports from nine institutions. There were no differences in demographics between subject cohorts completing the dietary assessment and blood analysis portions of the study except that the blood cohort included 10 different male sports and 11 different female sports, and the Pennsylvania institution did not participate in the blood analysis (Table 1). These cohorts are representative of NCAA Division I athletes across the U.S. in respect to sex (NCAA Division I = 53% male, diet sample = 51% male, blood sample = 55% male) and mean age (NCAA Division I = 20.3 years, diet sample = 19.9 years, blood sample = 20.0 years). In order to protect anonymity and confidentiality of our high profile population, we have not made full data points publicly available.

### Diet

Frequency of fish and seafood consumed and sources of fish and seafood consumed by participants are shown in Figs 1 and 2, respectively. A total of 601 participants (39%) met the recommendation to consume two or more fish servings weekly (Fig 1). Salmon and shrimp were the

**Table 1. Descriptive characteristics of participants.**

|  | Dietary Assessment | Blood Fatty Acid Analysis | Differences Test Statistic (p-value) |
|---|---|---|---|
| **n** | 1,528 | 298 |  |
| **Sex (Male/Female)** | *Male*: 780 (51.0%) | *Male*: 163(54.6%) | $\chi^2$ = 1.318/ (p = 0.251) |
|  | *Female*: 748 (49.0%) | *Female*: 135 (45.3%) |  |
| **Age (years; mean ± SD)** | 19.9 ± 1.4 | 20.0 ± 1.3 | F = 1.610/ (p = 0.646) |
| **Academic year n (%)** | *Freshman*: 442 (28.9%) | *Freshman*: 88 (29.5%) | $\chi^2$ = 18.50/ (p = 0.470) |
|  | *Sophomore*: 373 (24.4%) | *Sophomore*: 73 (24.4%) |  |
|  | *Junior*: 377 (24.7%) | *Junior*: 70 (23.5%) |  |
|  | *Senior*: 270 (17.7%) | *Senior*: 58 (19.4%) |  |
|  | *5th year or Graduate*: 63 (4.1%) | *5th year or Graduate*: 9 (3.0%) |  |
| **Sport n (%)** | [a] *Male Sports*: 780 (51.0%) | [c] *Male Sports*: 163 (54.6%) | $\chi^2$ = 4.779/ (p = .1912) |
|  | *Football*: 303 (19.8%) | *Football*: 81 (27.2%) |  |
|  | *Non-football*: 477 (31.2%) | *Non-football*: 82 (27.5%) |  |
|  | [b] *Female Sports*: 748 (49.0%) | [d] *Female Sports*: 115 (45.3%) |  |
| **Region n (%)** | *California*: 106 (6.9%) | *California*: 28 (28.6%) | $\chi^2$ = 7.003/ (p = .0991) |
|  | *Georgia*: 158 (10.3%) | *Georgia*: 33 (11.1%) |  |
|  | *Illinois*: 77 (5.0%) | *Illinois*: 29 (9.7%) |  |
|  | *Nebraska*: 211 (13.8%) | *Nebraska*: 45 (15.1%) |  |
|  | *Oregon*: 111 (7.3%) | *Oregon*: 39 (13.1%) |  |
|  | *Pennsylvania*: 61 (4.0%) | *Pennsylvania*: 0 (0.0%) |  |
|  | *Texas*: 336 (22.0%) | *Texas*: 40 (13.4%) |  |
|  | *Utah*: 102 (6.7%) | *Utah*: 42 (14.1%) |  |
|  | *Virginia*: 365 (23.9%) | *Virginia*: 43 (14.4%) |  |

[a] Baseball, Basketball, Cross Country, Fencing, Football, Golf, Gymnastics, Ice Hockey, Ski, Soccer, Swimming & Diving, Tennis, Track & Field, Volleyball, Wrestling

[b] Basketball, Beach Volleyball, Bowling, Cross Country, Equestrian, Fencing, Field Hockey, Golf, Gymnastics, Lacrosse, Rifle, Rowing, Ski, Soccer, Softball, Swimming & Diving, Tennis, Track & Field, Volleyball

[c] Baseball, Basketball, Cross Country, Football, Golf, Gymnastics, Tennis, Track & Field, Wrestling

[d] Basketball, Cross Country, Fencing, Gymnastics, Lacrosse, Rifle, Rowing, Soccer, Softball, Swimming & Diving, Track & Field

only EPA and DHA sources reported to be consumed by more than 50% of participants (Fig 2). Sources of ALA included canola oil (85%), walnuts (53.9%), chia (43.6%), flax or flax oil (34.9%), and cod liver oil (3.3%).

Male participants consumed significantly more EPA and DHA than female participants and female participants consumed significantly more ALA than male participants (Table 2). Only 91 (6%) of participants consumed at least 500 mg combined EPA + DHA/day as advised by the Academy of Nutrition and Dietetics [33] and 62 (4%) met the National Academy of Medicine's recommendation of 1.6 g ALA (men) and 1.1 g ALA (women) [35]. There were no significant differences found in EPA, DHA, ALA, EPA + DHA or total omega-3 intake when comparing individual sports with greater than 8 participants (men's and women's basketball, baseball, football, men's and women's soccer, men's and women's swim and dive, men's track & field, wrestling and softball, women's golf, women's gymnastics, women's lacrosse and women's volleyball).

Use of ω-3 FA supplements was reported by 229 participants (15%). A total of 153 (10%) purchased the supplement on their own, while 76 (5%) received supplements from their respective athletic program via physician prescription. Of the nine participating programs, seven were reported as providing supplements and of the 76 participants receiving supplements, 49 (69%) were football athletes. Most participants provided no response to brand, type, and dose of ω-3 FA supplements consumed.

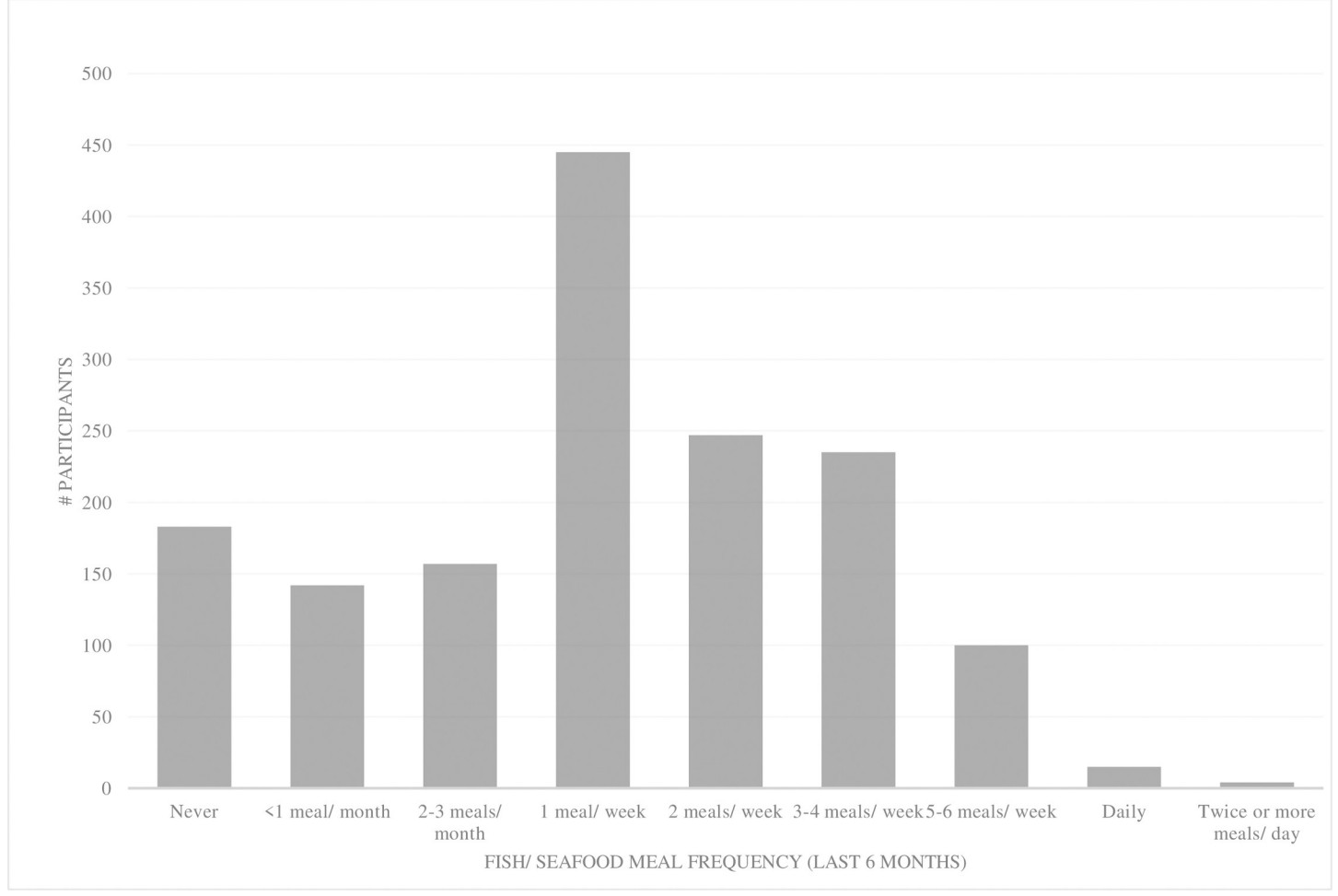

**Fig 1. Frequency of fish and seafood consumption during the previous 6 months (n = 1528).**

### Blood

Result of blood fatty acid and O3i analyses are shown in Table 3. O3i ranged from 2.25 to 7.23% (Fig 3), with 114 (38%) in the high risk category, 184 (62%) in the moderate risk category, and 0 (0%) in the low risk category (Fig 3). There were no significant differences in blood measures based on sex (Fig 4), institution, age, or academic year. There was no significant difference found in O3i when comparing individual sports with greater than 8 participants (men's and women's basketball, baseball, football, men's and women's soccer, women's swim and dive, men's track & field, wrestling and softball, women's gymnastics, women's lacrosse and women's volleyball).

### Relationship between diet and blood measures

Dietary intake of both EPA and DHA were positively correlated with blood EPA, DHA, and O3i (Table 4). There was no correlation between dietary ALA intake and blood levels of EPA, DHA, ALA or O3i (Table 4).

   After controlling for institution, sex, age, class year and sport (football vs. non-football), frequency of fish and seafood consumption was a significant predictor of O3i ($R^2 = .3701$, $p<0.01$). Each additional serving of seafood was associated with a O3i increase of 0.27% (Fig

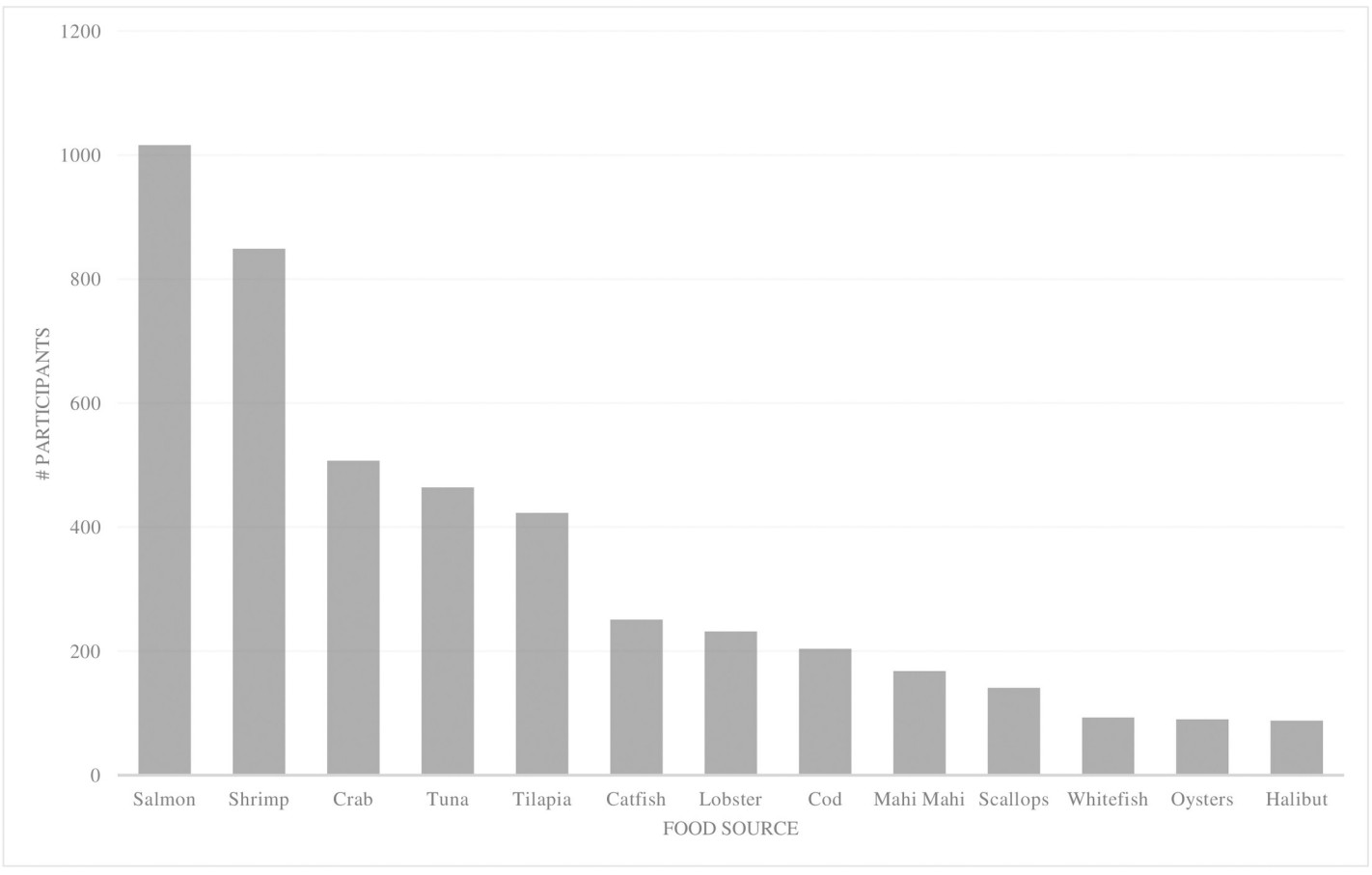

**Fig 2. Sources of fish and seafood consumed during the previous 6 months (n = 1528).**

5). Participants who reported taking ω-3 FA supplements had significantly higher O3i compared with those not taking supplements (4.7 vs. 3.7%, respectively; $p<0.05$). Participants who met the Academy of Nutrition and Dietetics' recommendation of 500 mg EPA + DHA per day had a higher O3i compared to those who consumed less than the daily 500 mg EPA + DHA recommendation (5.4% vs. 4.3%, $p<0.05$).

## Discussion

The primary goal of this study was to describe the ω-3 FA status of NCAA Division I athletes in the U.S. Our findings indicate that most NCAA Division I athletes do not meet current

**Table 2. Dietary consumption of Omega-3 fatty acids (n = 1528).**

|  | Total Daily Intake (mg) | Sex | | p-value |
|---|---|---|---|---|
|  |  | Male n = 780 | Female n = 748 |  |
| EPA | 46.8 +/- 86.9 | 53.4 | 40.4 | .0042** |
| DHA | 94.8 +/- 164.9 | 106.4 | 83.9 | .0091** |
| ALA | 571.8 +/- 1151.5 | 530.4 | 626.6 | .0281* |
| EPA + DHA | 141.7 +/- 250.6 | 159.8 | 124.3 | .0068** |

** $p<0.01$

**Table 3. Blood fatty acid analysis results (n = 298).**

| | Blood Fatty Acids (%) | Sex | | p-value |
|---|---|---|---|---|
| | | Male | Female | |
| EPA | 0.45 ± 0.19 | 53.4 | 40.4 | p = 0.704 |
| DHA | 2.19 ± 0.59 | 106.4 | 83.9 | p = 0.699 |
| ALA | 0.49 ± 0.19 | 530.4 | 626.6 | p = 0.588 |
| O3i | 4.3 ± 0.81 | 4.3 | 4.4 | p = 0.905 |

dietary recommendations for ω-3 FA and have sub-optimal O3i as compared to currently proposed cardiovascular benchmarks. To our knowledge, this is the first large scale assessment of ω-3 FA status of male and female collegiate athletes from a variety of sports.

Given the pattern of inadequate ω-3 FA status observed among NCAA Division I athletes, clinicians should consider nutritional interventions aimed at improving ω-3 FA status. One strategy could be increasing consumption of fish and seafood, the richest sources of EPA and DHA, as nearly half of participants reported no fish consumption in the last 6 months. Based on our findings, more frequent inclusion of ω-3 FA-rich sources in provided meals is an encouraged method for improving these low intakes. Capitalizing on popular fish and seafood sources (salmon, shrimp, crab, tuna, and tilapia were consumed the most in the current study)

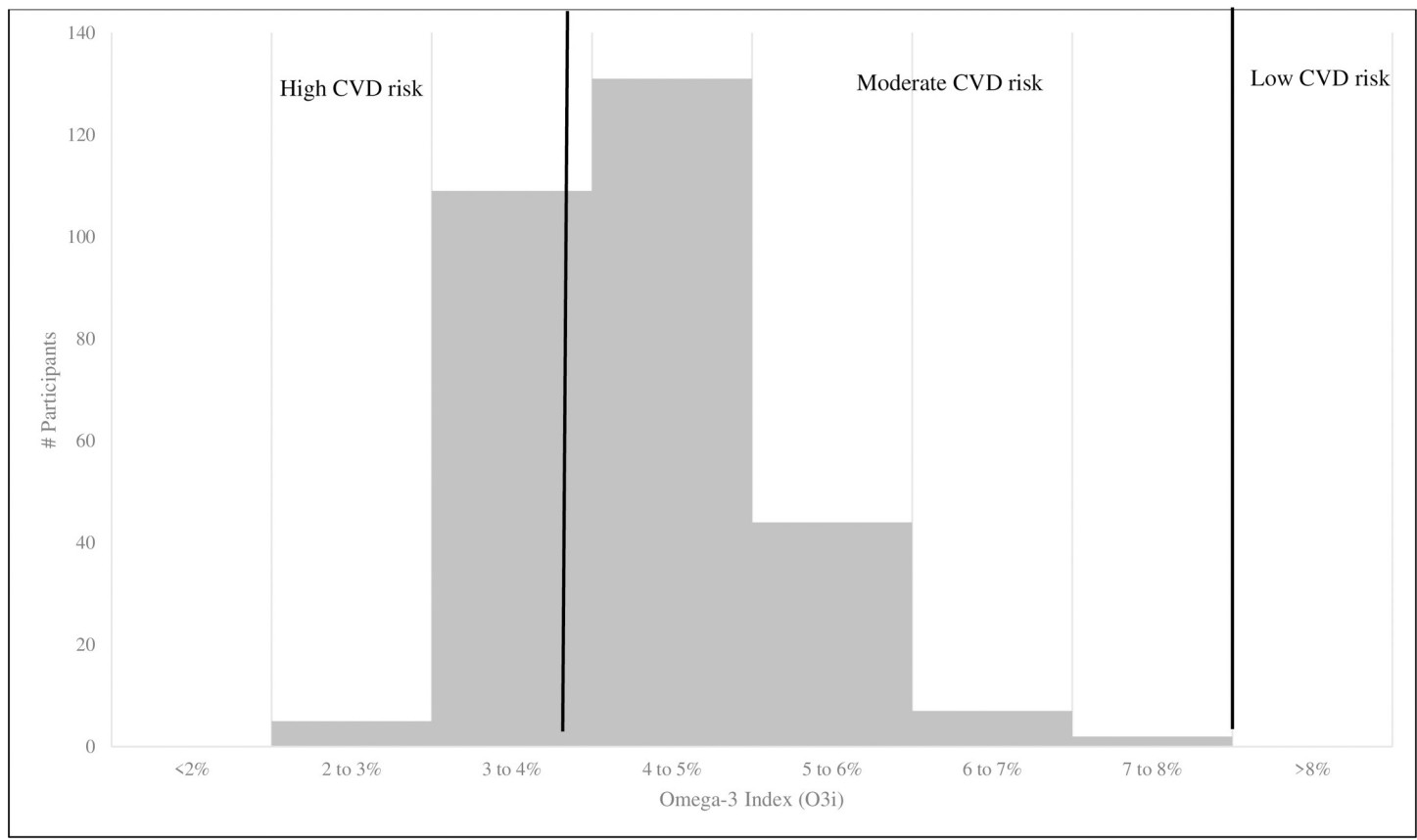

Note. Ranges associated with risk for development of cardiovascular disease. [26,31,32]

**Fig 3. Distribution of Omega-3 index results (n = 298).** Ranges associated with risk for development of cardiovascular disease [4–6].

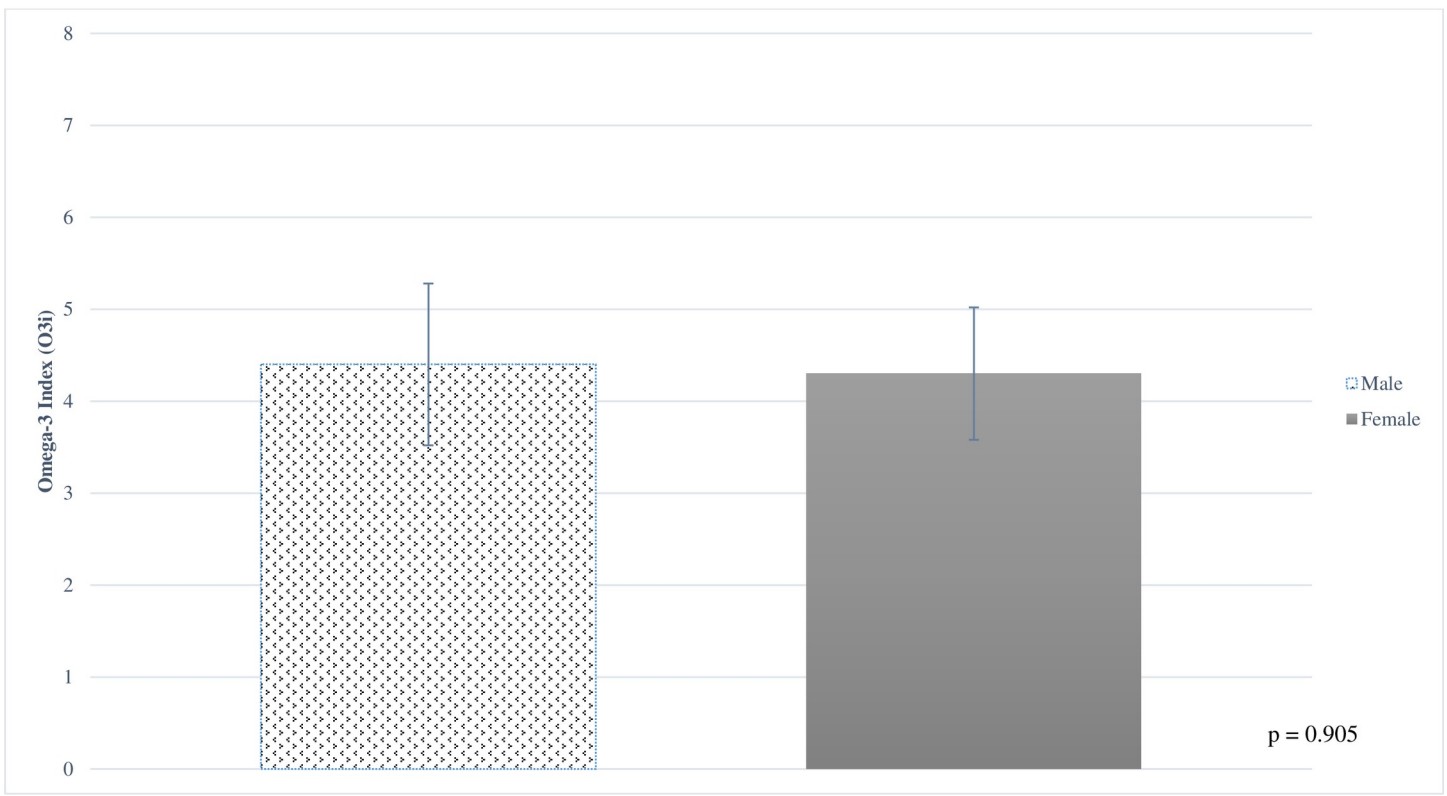

**Fig 4. Omega-3 index in male and female NCAA division 1 student athletes.**

may be beneficial [49]. However, those involved in nutrition programming and meal planning should consider the independent omega-3 profiles of these foods. Emphasizing the richest sources of DHA and EPA by incorporating sources like salmon, trout and mackerel, for example, provide a more concentrated dose aimed at improving O3i status. Practitioners should also recognize that plant-based sources of ω-3 FA are only rich in ALA and that the conversion of ALA to EPA and DHA is minimal [27]. The observed lack of correlation between dietary ALA and blood measures of EPA, DHA and O3i in the current study, is also consistent with previous findings [39, 47].

**Table 4. Diet and blood fatty acid correlations table.**

| | Diet EPA | Diet DHA | Diet EPA + DHA | Diet ALA | Diet Total ω-3 | Blood EPA | Blood DHA | Blood ALA | Blood O3i |
|---|---|---|---|---|---|---|---|---|---|
| **Diet EPA** | 1 | | | | | | | | |
| **Diet DHA** | .977 ** | 1 | | | | | | | |
| **Diet EPA + DHA** | 0.990 ** | 0.997 ** | 1 | | | | | | |
| **Diet ALA** | 0.134 | 0.154 | 0.148 | 1 | | | | | |
| **Diet Total ω-3** | 0.332 * | 0.552 * | 0.347 * | 0.979 ** | 1 | | | | |
| **Blood EPA** | 0.342 * | 0.334 * | 0.338 * | 0.296 | 0.339 * | 1 | | | |
| **Blood DHA** | 0.397 * | 0.404 * | 0.403 * | 0.214 | .273 | .402 * | 1 | | |
| **Blood ALA** | 0.072 | 0.080 | 0.078 | 0.090 | 0.098 | 0.072 | -0.122 | 1 | |
| **Blood O3i** | 0.437 * | 0.441 * | 0.442 * | 0.271 | 0.332 * | 0.648 ** | 0.958 ** | -0.079 | 1 |

*p<0.05,

**p<0.01

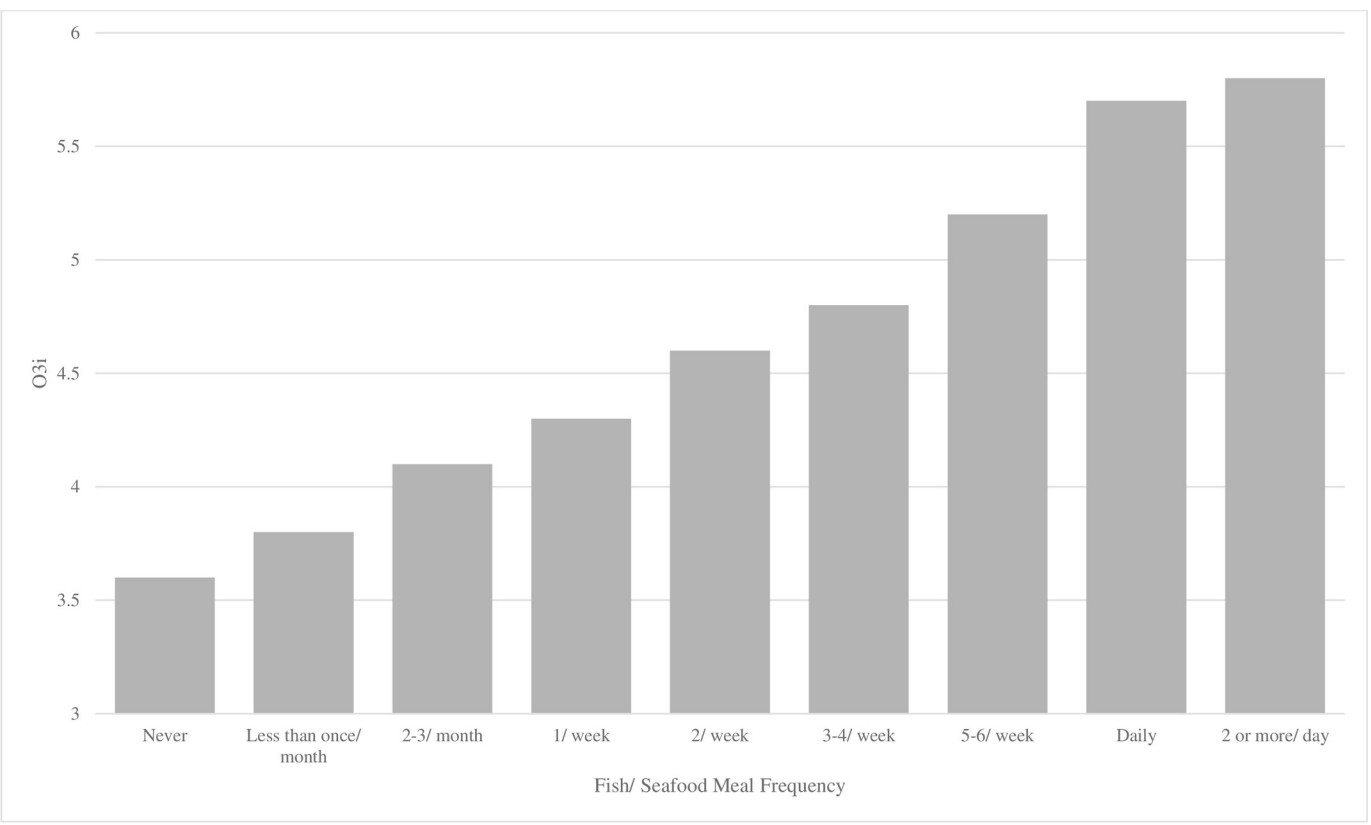

**Fig 5. Association between fish or seafood meal frequency and omege-3 index (n = 298).** After controlling for institution, sex, age, class year and sport (football vs. non-football), frequency of fish and seafood consumption was a significant predictor of O3i ($R^2$ = .3701, p<0.01). Each additional serving of seafood was associated with a O3i increase of 0.27%.

In recent years, the NCAA has seen significant changes in terms of the feeding opportunities available for Division I athletes as a result of the deregulation of meal restrictions in 2014, allowing institutions more flexibility in the provision of nutrition to athletes [50]. It is important to note that this legislation change was specific to Division I athletes, and Division II and III (in addition to non-Power Division I) athletes may face greater obstacles in terms of meeting adequate intake as a result of each program's unique financial limitations.

Although the majority of collegiate athletes participating in the present study did not meet current dietary ω-3 FA recommendations—similar to previous observations [39]—these guidelines are not specific to athletes. Further research is needed to establish athlete-specific recommendations, taking into consideration the physiological implications of advanced levels of training on metabolism and the inflammatory response [51–53]. For example, lower average O3i was observed among non-elite runners with greater training mileage compared to those with lesser running mileage [51]. Additional research is also needed to identify intake of ω-3 FA most effective for neuroprotection and brain health.

No participant in the current study, including those who consumed fish or seafood twice or more per week, had an O3i of 8%, the level associated with lowest cardiovascular disease risk [4–6]. Thus, achieving optimal ω-3 FA status through diet alone may be difficult and it is plausible that athletes may actually have higher needs than the general population. The use of ω-3 FA supplements is another strategy for improving ω-3 FA status, and has been discussed as a potentially helpful nutritional tool for athletes [54]. A small percentage of participants reported

ω-3 FA supplement use but almost none were able to provide information about brand, form, dosage, and frequency of supplements used. The recent NCAA guidelines changes [55] present an opportunity to more readily provide ω-3 FA when appropriate, and to do so in a safe, controlled, and monitored fashion.

The sub-optimal O3i observed for in our study (4.3%) was similar to previous observations [39, 44, 56, 57], and did not differ based on sex or sport. Football-specific findings in this study were virtually identical to previous findings in Division I football athletes with both groups averaging an O3i of 4.4% [44]. While further research is needed to investigate potential differences in needs between athletes of different sex and sport, we observed NCAA Division I athletes collectively have low ω-3 FA status. Interestingly, the higher consumption of EPA and DHA observed in male participants compared to females did not translate to higher O3i values. This might suggest external factors such as higher average body mass, higher caloric needs and availability of athletic department nutrition resources drove the observed increases in EPA and DHA intake and was not significant enough to impact blood status. To our knowledge, no U.S.-based athletes have been documented in the peer reviewed literature as having O3i greater than 8% [39, 44], the proposed benchmark for optimal cardiovascular health [4–6]. Research suggests EPA and DHA may reduce cardiovascular risk factors such as dyslipidemia and high blood pressure[58–60] and even sudden death related to cardiac causes [61–65], which are applicable to a wide variety of athletes. This will continue to be of particular interest for higher mass athletes (such as linemen in American football) observed to be at higher risk of cardiovascular disease and metabolic syndrome [61, 66, 67]. Given this concern, in combination with the associated health functions of ω-3 FA related to both athlete performance and well-being [2, 3, 8–13, 15–24, 26], a focus on improved O3i is warranted. It is important to note, however, that target O3i for non-cardiovascular conditions is not well-established and continuing research is needed to investigate the impact of O3i on athlete health and performance measures.

## Strengths & limitations

Collaboration with a diverse group of Power 5 institutions enabled us to study a large sample of athletes from nearly every NCAA sport with varying dietary habits and available resources. Further, given the timing of the NCAA legislation changes in relation to the timeline of our assessment, this investigation serves as a baseline for ω-3 FA intake and ω-3 FA supplement use among NCAA Division I athletes. Finally, our results parallel those of others who have observed a positive correlation between dietary EPA and DHA intake and O3i [68–70]. This suggests that the FFQ used was a reliable measure of ω-3 FA intake [39, 47]. This FFQ provides a cost-effective method for assessing ω-3 FA status in clinical situations where blood assessment may not be practically or financially warranted.

The study does have some limitations, however. In an effort to recruit a large, geographically-diverse cohort and provide equitable experiences for all participants, we included athletes from all sports sponsored by each institution in the study. Although we achieved a large and very diverse cohort (with male/female representation similar to that of the overall NCAA Division 1 membership), there was variability in number of participants from each sport and it was difficult to make sport by sport comparisons. With this in mind, we categorized sports as football vs. non-football for many analyses in order to compare our data with other published results [44]. As there are characteristics that often distinguish football from other intercollegiate sports, such as programmatic resources, athlete size, and occurrence of head injury, we believe this categorization is relevant. It should be acknowledged that while participating institutions were assured anonymity, inclusion of state and Power 5 criteria does allow the reader

to make some assumptions about the identity of institutions. Although we believe that our results are generalizable to NCAA Division I athletes, further sports-specific investigations would be valuable, as would evaluation of Division II and III athletes.

Regarding the dietary assessment, fish and seafood vary in nutritional content based on a number of factors, including variety consumed, location, and time of year. Our assessment did not account for this variation. Additionally, we did not collect data related to race/ethnicity, height, and body weight in effort to assure anonymity of participants, but this information may have been insightful in data analysis. Overall, the lack of universally accepted dietary recommendations and blood measure standards provided an additional obstacle in terms of interpreting our results, which should be a primary motive for future research.

## Conclusion

Prior to the 2019 change in NCAA legislation change related to ω-3 FA supplementation, we observed sub-optimal omega-3 status in NCAA Division I athletes based on both dietary and blood assessments. These results serve to inform future nutritional interventions aimed at improving ω-3 FA status among athletes. Results also provide a baseline in order to measure the impact of nutrition interventions created as a result of this legislation change.

## Acknowledgments

The authors would like to thank the research collaborators at the nine participating institutions for their dedication to the project. This study was supported by the Collegiate and Professional Sports Dietitians Association Research Award. There was no additional external funding received for this study.

## Author Contributions

**Conceptualization:** Peter P. Ritz, Mark B. Rogers, Jennifer S. Zabinsky, Matthew W. Hulver, Michelle S. Rockwell.

**Data curation:** Peter P. Ritz, Valisa E. Hedrick, John A. Rockwell, Samantha B. Kostelnik, Michelle S. Rockwell.

**Formal analysis:** Peter P. Ritz, Valisa E. Hedrick, John A. Rockwell, Michelle S. Rockwell.

**Funding acquisition:** Peter P. Ritz, Michelle S. Rockwell.

**Investigation:** Peter P. Ritz, Ernest G. Rimer, Michelle S. Rockwell.

**Methodology:** Peter P. Ritz, Jennifer S. Zabinsky, John A. Rockwell, Michelle S. Rockwell.

**Project administration:** Peter P. Ritz, Jennifer S. Zabinsky, Ernest G. Rimer, Michelle S. Rockwell.

**Resources:** Michelle S. Rockwell.

**Supervision:** Michelle S. Rockwell.

**Visualization:** Peter P. Ritz, Valisa E. Hedrick.

**Writing – original draft:** Peter P. Ritz.

**Writing – review & editing:** Peter P. Ritz, Mark B. Rogers, Jennifer S. Zabinsky, Valisa E. Hedrick, John A. Rockwell, Ernest G. Rimer, Samantha B. Kostelnik, Matthew W. Hulver, Michelle S. Rockwell.

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
