## [Decision Letter · Decision Letter 0]

12 Feb 2020

PONE-D-20-02046

Dietary and Biological Assessment of Omega-3 Status of Collegiate Athletes: A Cross-Sectional Analysis

PLOS ONE

Dear Dr. Rockwell,

Thank you for submitting your manuscript to PLOS ONE. After careful consideration, we feel that it has merit but does not fully meet PLOS ONE’s publication criteria as it currently stands. Therefore, we invite you to submit a revised version of the manuscript that addresses the points raised during the review process.

While completing revisions, I encourage you to consider the following:

Inclusion of tables in the introductionJustification for the population used in your study Clarity of the methodology and testing procedures

We would appreciate receiving your revised manuscript by March 10, 2020. To enhance the reproducibility of your results, we recommend that if applicable you deposit your laboratory protocols in protocols.io, where a protocol can be assigned its own identifier (DOI) such that it can be cited independently in the future. For instructions see: http://journals.plos.org/plosone/s/submission-guidelines#loc-laboratory-protocols

We look forward to receiving your revised manuscript.

Kind regards,

Emily Sauers

Academic Editor

PLOS ONE

2. In your Methods section, please provide additional information on how the participating institutions were selected. Moreover, please ensure you have provided sufficient details to replicate the analyses such as: a)  a statement as to whether your sample can be considered representative of a larger population, b) a description of how participants were recruited, and c) descriptions of where participants were recruited and where the research took place.

3. Thank you for stating in your Acknowledgements Section:

"This study was funded, in part, by the Collegiate and Professional Sports Dietitians Association Research Award."

4. Please ensure that you refer to Figure 6 in your text as, if accepted, production will need this reference to link the reader to the figure.

Reviewers' comments:

Reviewer's Responses to Questions

**Comments to the Author**

1. Is the manuscript technically sound, and do the data support the conclusions?

Reviewer #1: Partly

Reviewer #2: Yes

2. Has the statistical analysis been performed appropriately and rigorously? 

Reviewer #1: Yes

Reviewer #2: Yes

3. Have the authors made all data underlying the findings in their manuscript fully available?

Reviewer #1: No

Reviewer #2: Yes

4. Is the manuscript presented in an intelligible fashion and written in standard English?

Reviewer #1: Yes

Reviewer #2: Yes

5. Review Comments to the Author

Reviewer #1: PONE-D-20-02046 presents results from a sample of collegiate athletes. While some parts of this manuscript were interesting, other areas could be improved. I hope the authors consider my feedback for enhancing their manuscript.

MAJOR COMMENTS

• Introduction: It is generally rare to insert tables such as these in Introduction sections (probably more so relevant for thesis and dissertation papers). A lot of this information could be abbreviated and inserted where appropriate in the text. While I understand the information the authors are trying to convey here, it just comes across as odd and distracting to this reviewer.

• Participants: How were the institutions selected within each state, and why were these specific states selected (e.g., Illinois vs. Minnesota)? Can specific institutions also be identified?

• Methods: More information about sampling would be helpful. For example, was a snowball sampling technique used? How were missing data on questionnaires treated? How were questionnaire administered (e.g., in-person)?

• Statistical analysis: More detail about the statistical analyses and how it is compatible with what is presented in the results is needed. For example, what specific demographic covariates were in the multiple regression analysis?

• Results: Why were data dichotomized by football for most tables? By doing this, you are also merging sexes in the non-football category. This could be problematic because there appears to be significant differences between sexes (e.g., Table 4).

MINOR COMMENTS

• Line 5: Be sure to use consistency with in-text citations in the manuscript “,(1-6)” vs “(14-18),”.

• Table 1: Consider instead listing frequency and percent for sex, similar to what was done for academic year, etc.

• Methods: Can you confirm in the manuscript that each participating institution had the same sports offered across institutions?

• Figures 4 and 5: Please insert measures of variability in the figure.

• Figure 6 may need a new title. “Relationship” suggests there was a statistical test performed. The figure does not represent this in its current form.

• Discussion: Could the authors speculate how these findings would generalize to non-power 5 athletes and athletes in Divisions II and III?

Reviewer #2: The objective of this study was to characterize dietary intake of omega-3’s and determine the omega-3 index, a measure of EPA and DHA in red blood cell membranes, in collegiate athletes. Similar to the general population, very few athletes met recommendations for diet or the omega-3 index. These findings are interesting as the NCAA recently (2019) began permitting the use of omega-3 supplements. Discussion on why the NCAA reclassified omega-3’s would be helpful. Additionally, the authors discuss their findings in the context of cardiovascular disease, but this should be justified, as I suspect collegiate athletes are a lower risk of cardiovascular disease compared to the general population. Additional suggestions are included below:

• Was there a rationale for the football, non-football male sport, and female sport grouping? It may be worth mentioning the relationship between traumatic brain injury and omega-3 supplementation.

• The authors discuss the omega-3 index in terms of cardiovascular disease, which I assume is low in this population (collegiate athletes) compared to the general population. Please justify.

• Line 17: Remove table 1, provide a reference (USDA nutrient database), and consider listing the range of EPA/DHA in the context of commonly consumed choices.

• Line 33: Remove table 2 and include key references in the text body.

• Line 87: Were all participants invited to participate in the second phase of the study? If not, were the invitations to participate randomized?

• Line 129: Consider, “601 participants (39%) consumed at least two or more servings of fish weekly.”

• Line 137: If available, it may be worth noting which teams provided supplements vs. which teams did not.

• Line 165: Should this be figure 6, not figure 5?

• Line 218: reference (55) is cardiovascular risk of American football athletes not all. The review article (55) that is cited says “This review was constructed to delineate our contemporary understanding of cardiovascular health among American-Style Football participants”.

• Figures 4-6 should include error bars and statistical analyses should be conducted.

6. PLOS authors have the option to publish the peer review history of their article (what does this mean?). If published, this will include your full peer review and any attached files.

Reviewer #1: No

Reviewer #2: No

---

## [Author Response · Author response to Decision Letter 0]

10 Mar 2020

Additional Requirements

 Feedback Author Response

1 Please ensure that your manuscript meets PLOS ONE's style requirements, including those for file naming. We have reviewed the manuscript in detail to meet PLOS ONE’s style requirements. Multiple revisions were made, particularly in the reference section. 

2 In your Methods section, please provide additional information on how the participating institutions were selected. 

Moreover, please ensure you have provided sufficient details to replicate the analyses such as: a) a statement as to whether your sample can be considered representative of a larger population, b) a description of how participants were recruited, and c) descriptions of where participants were recruited and where the research took place.

 Thank you for the opportunity to clarify the methodology used for the study. Several edits have been made within the Methods section, including the requested details. 

Of note, our study cohort represents almost 1% (1528/≈160,000) of the NCAA Division 1 student-athlete population and 5% (1528/≈30,000)of the NCAA Division 1 Power 5 student-athlete population. Institutions from throughout the U.S. were included among our study cohort (intentionally representing the 8 geographic regions), our gender breakdown matches that of the NCAA Division 1 student-athlete population, and almost every NCAA Division 1 sport was included among our cohort. Thus, we feel that our cohort is a strong representation of the population. However, we readily acknowledge that different sampling methodology may have led to a more perfectly representative cohort. We have added this as a limitation of the study. 

3 Thank you for stating in your Acknowledgements Section:

"This study was funded, in part, by the Collegiate and Professional Sports Dietitians Association Research Award."

 We have modified the funding statement to indicate that the only funding received for the study was from the CPSDA Research Award, and that no additional external funding was received. The “in part” phrase is incorrect!

4 Please ensure that you refer to Figure 6 in your text as, if accepted, production will need this reference to link the reader to the figure. Thank you for your catching this typo. This figure is now properly labeled within the text. 

5 The PLOS Data policy requires authors to make all data underlying the findings described in their manuscript fully available without restriction, with rare exception (please refer to the Data Availability Statement in the manuscript PDF file). The data should be provided as part of the manuscript or its supporting information, or deposited to a public repository. For example, in addition to summary statistics, the data points behind means, medians and variance measures should be available. If there are restrictions on publicly sharing data—e.g. participant privacy or use of data from a third party—those must be specified.

 While we appreciate and support PLOS ONE’s commitment to data transparency, we have not made full data tables available on a public repository. This is because we assured participating institutions anonymity for themselves and their student-athletes who are considered a high profile, sensitive population. A statement to this effect was added to the results section in the manuscript. 

We are willing to discuss this issue further with the editor, reviewers, and even the IRB and participating institutions if necessary. 

Reviewer 1

 Feedback Author Response

MAJOR COMMENT 1 Introduction: It is generally rare to insert tables such as these in Introduction sections (probably more so relevant for thesis and dissertation papers). A lot of this information could be abbreviated and inserted where appropriate in the text. While I understand the information the authors are trying to convey here, it just comes across as odd and distracting to this reviewer. We have removed tables from the introduction and incorporated key abbreviated information int the text. This edit does improve the flow and readability of the introduction – thank you. 

MAJOR COMMENT 2 Participants: How were the institutions selected within each state, and why were these specific states selected (e.g., Illinois vs. Minnesota)? Can specific institutions also be identified? We have clarified institution recruitment within the text; thank you for this recommendation. Briefly, we solicited volunteer institutions from within the NCAA Division I Power 5 membership. On a first come/first served basis, we accepted one institution from each of eight pre-defined regions within the U.S. (Northwest, Southwest, etc.). We ultimately accepted two Northeast programs since the first became unable to collect blood samples due to a staffing change. 

Prior to the start of the study, participating institutions were promised protection of program identity at the level of state. Thus, programs were identified only by state (and NCAA Division I Power 5). 

MAJOR COMMENT 3 Methods: More information about sampling would be helpful. For example, was a snowball sampling technique used? How were missing data on questionnaires treated? How were questionnaire administered (e.g., in-person)? Additional details have been added throughout the methods section. For example, we have noted that student-athletes volunteered to participate through convenience sampling, which included open recruitment of all student-athletes at each institution (flyers in common/shared athletics facilities, email, and text messages). Questionnaires were administered electronically and responses with more than one question left blank were excluded from analyses (n=34). 

MAJOR COMMENT 4 Statistical analysis: More detail about the statistical analyses and how it is compatible with what is presented in the results is needed. For example, what specific demographic covariates were in the multiple regression analysis? We have clarified language around the statistical analyses we used. Demographic covariates (institution, sex, age, and class year, and sport) were added to the text. 

MAJOR COMMENT 5 Reviewer 1) Why were data dichotomized by football for most tables? By doing this, you are also merging sexes in the non-football category. This could be problematic because there appears to be significant differences between sexes (e.g., Table 4).

 The research team revisited this important issue and made several modifications. Realizing that our study design was likely to yield a large number of total athletes (but not necessarily a large or equivalent number of student-athletes from each different sport), our study plan included analysis of football data vs. other sport data. This would allow for comparison of our results to the existing literature, which primarily involves football (ex: Anzalone et al., 2019). Furthermore, other characteristics commonly distinguish football from other intercollegiate sports, including programmatic resources (budget, meals, supplements, staff, etc.), athlete size, and occurrence of head injury (though we acknowledge that head injury is an issue for many athletes from many different sports). 

However, upon re-examining the results, we decided to keep some football vs. non-football comparisons, but to minimize these throughout the manuscript (removing it from the tables). Although the significant difference in football vs. non-football athletes remained in EPA + DHA when controlling for sex (regression analysis), we realize that the comparison may not be ideal since data on resource differences, athlete size, etc. were not collected.

If the reviewer would like to see additional changes made to our analysis and interpretation, we welcome the option to consider. 

MINOR COMMENT 1 Be sure to use consistency with in-text citations in the manuscript “,(1-6)” vs “(14-18),”. Citation formatting has been edited to consistently reflect PLOS ONE’s formatting requirements. We apologize for neglecting this important detail in the initially submitted manuscript. 

MINOR COMMENT 2 Table 1: Consider instead listing frequency and percent for sex, similar to what was done for academic year, etc. This edit has been made in Table 1.

MINOR COMMENT 3 Methods: Can you confirm in the manuscript that each participating institution had the same sports offered across institutions? Unfortunately, each participating institution did not offer all of the same sports, but there was a great deal of overlap. All nine institutions offer M baseball, M & W basketball, M & W cross country, M football, M & W golf, W gymnastics, W soccer, W softball, M & W swimming & diving, M & W track & field, and W volleyball. We considered including only student-athletes from these sports. However, since our hypothesis was that there would be no difference in ω-3 FA status based on sport, we elected to invite student-athletes from all teams to participate in this cross sectional study (which helped to facilitate a large sample size). Our results indicate that differences in ω-3 FA status seem to be based on gender rather than sport. Additionally, research collaborators and the research team felt that inviting all athletes on all teams to participate was presented the best equitability to student-athletes, which is a priority among NCAA institutions. 

We have added language to the Methods section indicating that there were some differences in sports offered at each institution and also added this issue as a study limitation. 

MINOR COMMENT 4 Figures 4 and 5: Please insert measures of variability in the figure.

 Measures of variability and error bars have been added. Thank you for this important recommendation. 

MINOR COMMENT 5 Figure 6 may need a new title. “Relationship” suggests there was a statistical test performed. The figure does not represent this in its current form.

 We have edited the title of the figure to refer to Association rather than Relationship. We also added a footnote to refer to the results of the regression analysis conducted on the fish/seafood frequency and O3i data. 

MINOR COMMENT 6 Discussion: Could the authors speculate how these findings would generalize to non-power 5 athletes and athletes in Divisions II and III? This is an important recommendation, which we appreciate. Information about generalizability of our results was added to the discussion. 

 

Reviewer 2

 Feedback Author Response

GENERAL REMARKS Discussion on why the NCAA reclassified omega-3’s would be helpful.

 A statement was added to the introduction related to this remark. Member institution advocacy (motivated by health and safety benefits of ω-3 FA) led to the NCAA reclassification.

COMMENT 1 Was there a rationale for the football, non-football male sport, and female sport grouping? It may be worth mentioning the relationship between traumatic brain injury and omega-3 supplementation. 

 Thank you for the important question and suggestion. We have added mention of the relationship between brain health and ω-3 FA. 

Additionally, the research team revisited the football vs. non-football classifications and made several modifications. Realizing that our study design was likely to yield a large number of total athletes (but not necessarily a large or equivalent number of student-athletes from each different sport), our study plan included analysis of football data vs. other sport data. This would allow for comparison of our results to the existing literature, which primarily involves football (ex: Anzalone et al., 2019). Furthermore, other characteristics commonly distinguish football from other intercollegiate sports, including programmatic resources (budget, meals, supplements, staff, etc.), athlete body size, and occurrence of head injury (though we acknowledge that head injury is an issue for many athletes from many different sports). 

However, upon re-examining the results, we decided to keep some football vs. non-football comparisons, but to minimize these throughout the manuscript (including removing it from the tables). Although the significant difference in football vs. non-football athletes remained in EPA + DHA when controlling for sex (regression analysis), we realize that the comparison may not be ideal since data on resource differences, athlete size, etc. were not collected.

If the reviewer would like to see additional changes made to our analysis and interpretation, we welcome the option to consider. 

COMMENT 2 The authors discuss the omega-3 index in terms of cardiovascular disease, which I assume is low in this population (collegiate athletes) compared to the general population. Please justify.

 Although O3i reflects the ω-3 FA content of a variety of tissues, a strong association between, O3i is only strongly associated with cardiovascular disease risk in the literature. 

The risk of cardiovascular disease and metabolic syndrome is elevated in some college athletes (such as football) and the risk of sudden cardiac death is elevated in many sports, which is why our O3i results are especially important.

Modifications have been made to the manuscript to reflect these points. 

COMMENTS 3 & 4 Introduction: Remove table 1, provide a reference (USDA nutrient database), and consider listing the range of EPA/DHA in the context of commonly consumed choices.

Introduction: Remove table 2 and include key references in the text body. We have made the recommended edits. Removing the tables and adding a summary/references has made for a much more readable introduction. Thank you!

COMMENT 5 Methods: Were all participants invited to participate in the second phase of the study? If not, were the invitations to participate randomized? The methods section has been edited to add more details, including information about recruitment. All participants who completed the dietary assessment were invited to participate in the blood analysis portion of the study. The exception to this is the Pennsylvania institution for whom blood analysis was not conducted due to staffing changes. 

COMMENT 6 Results: Consider, “601 participants (39%) consumed at least two or more servings of fish weekly.”

 This edit has been made. 

COMMENT 7 Results: If available, it may be worth noting which teams provided supplements vs. which teams did not This data is interesting, and we have added it to the result section. Of the supplements provided by athletic programs, 7 of 9 did provide supplements and 49 (69%) were to football athletes. 

Because of the existing NCAA legislation at the time of data collection, no institutions were able to provide ω-3 FA supplementation without a physician prescription.

COMMENT 8 Line 165: Should this be figure 6, not figure 5? Thank you for catching this typo. The statement has been corrected.

COMMENT 9 Line 218: reference (55) is cardiovascular risk of American football athletes not all. The review article (55) that is cited says “This review was constructed to delineate our contemporary understanding of cardiovascular health among American-Style Football participants”. We modified this statement to indicate football-specific observations, and also added additional references. 

COMMENT 10 Figures 4-6 should include error bars and statistical analyses should be conducted. These have been added where appropriate. Thank you.

---

## [Decision Letter · Decision Letter 1]

2 Apr 2020

PONE-D-20-02046R1

Dietary and Biological Assessment of Omega-3 Status of Collegiate Athletes: A Cross-Sectional Analysis

PLOS ONE

Dear Dr. Rockwell,

Thank you for submitting your manuscript to PLOS ONE. After careful consideration, we feel that it has merit but does not fully meet PLOS ONE’s publication criteria as it currently stands. Therefore, we invite you to submit a revised version of the manuscript that addresses the points raised during the review process.

Please address the following:

List each Power 5 Conference for clarity to readersExpand upon limitations

We would appreciate receiving your revised manuscript by April 15, 2020. To enhance the reproducibility of your results, we recommend that if applicable you deposit your laboratory protocols in protocols.io, where a protocol can be assigned its own identifier (DOI) such that it can be cited independently in the future. For instructions see: http://journals.plos.org/plosone/s/submission-guidelines#loc-laboratory-protocols

We look forward to receiving your revised manuscript.

Kind regards,

Emily Sauers

Academic Editor

PLOS ONE

Reviewers' comments:

Reviewer's Responses to Questions

**Comments to the Author**

1. If the authors have adequately addressed your comments raised in a previous round of review and you feel that this manuscript is now acceptable for publication, you may indicate that here to bypass the “Comments to the Author” section, enter your conflict of interest statement in the “Confidential to Editor” section, and submit your "Accept" recommendation.

Reviewer #1: (No Response)

Reviewer #2: All comments have been addressed

2. Is the manuscript technically sound, and do the data support the conclusions?

Reviewer #1: (No Response)

Reviewer #2: Yes

3. Has the statistical analysis been performed appropriately and rigorously? 

Reviewer #1: (No Response)

Reviewer #2: Yes

4. Have the authors made all data underlying the findings in their manuscript fully available?

Reviewer #1: (No Response)

Reviewer #2: No

5. Is the manuscript presented in an intelligible fashion and written in standard English?

Reviewer #1: (No Response)

Reviewer #2: Yes

6. Review Comments to the Author

Reviewer #1: • Please specifically list each Power 5 conference in the area in which it is first introduced to provide specificity to readers that may not know these conferences in a paratheatrical note.

• The authors indicate that participating institutions were assured protection of program identity; however, it is known that some states that only have one Power 5 institution will be identified by default (e.g., University of Utah). This should be listed as a limitation.

• The authors did a nice job of addressing the concern related to comparing football and non-football athletes in their revisions letter; however, the limitations of making these comparisons should also be listed in the limitations paragraph.

Reviewer #2: (No Response)

7. PLOS authors have the option to publish the peer review history of their article (what does this mean?). If published, this will include your full peer review and any attached files.

Reviewer #1: No

Reviewer #2: No

---

## [Author Response · Author response to Decision Letter 1]

3 Apr 2020

Reviewer #1: 

• Please specifically list each Power 5 conference in the area in which it is first introduced to provide specificity to readers that may not know these conferences in a paratheatrical note. This edit has been made.

• The authors indicate that participating institutions were assured protection of program identity; however, it is known that some states that only have one Power 5 institution will be identified by default (e.g., University of Utah). This should be listed as a limitation. This edit has been made.

• The authors did a nice job of addressing the concern related to comparing football and non-football athletes in their revisions letter; however, the limitations of making these comparisons should also be listed in the limitations paragraph. Thank you. This edit has also been made.

---

## [Editor Report · Decision Letter 2]

8 Apr 2020

Dietary and Biological Assessment of the Omega-3 Status of Collegiate Athletes: A Cross-Sectional Analysis

PONE-D-20-02046R2

Dear Dr. Rockwell,

We are pleased to inform you that your manuscript has been judged scientifically suitable for publication and will be formally accepted for publication once it complies with all outstanding technical requirements.

With kind regards,

Emily Sauers

Academic Editor

PLOS ONE

---

## [Editor Report · Acceptance letter]

17 Apr 2020

PONE-D-20-02046R2 

Dietary and Biological Assessment of the Omega-3 Status of Collegiate Athletes: A Cross-Sectional Analysis 

Dear Dr. Rockwell:

I am pleased to inform you that your manuscript has been deemed suitable for publication in PLOS ONE. Congratulations! Your manuscript is now with our production department. 

With kind regards,

on behalf of

Dr. Emily Sauers 

Academic Editor

PLOS ONE